# HSP70/IL-2 Treated NK Cells Effectively Cross the Blood Brain Barrier and Target Tumor Cells in a Rat Model of Induced Glioblastoma Multiforme (GBM)

**DOI:** 10.3390/ijms21072263

**Published:** 2020-03-25

**Authors:** Farzaneh Sharifzad, Soura Mardpour, Saeid Mardpour, Esmaeil Fakharian, Adeleh Taghikhani, Amirhossein Sharifzad, Sahar Kiani, Yasaman Heydarian, Marek J. Łos, Zahra Azizi, Saeid Ghavami, Amir Ali Hamidieh, Marzieh Ebrahimi

**Affiliations:** 1Department of Applied Cell Sciences, Kashan University of Medical Sciences, Kashan 87159-88141, Iran; fsharifzad@gmail.com (F.S.); efakharian@gmail.com (E.F.); 2Department of Stem Cells and Developmental Biology, Cell Science Research Center, Royan Institute for Stem Cell Biology and Technology, ACECR, Tehran 16635-148, Iran; souri_mardpour@yahoo.com (S.M.); adeleh80@yahoo.com (A.T.); Skiani2536@gmail.com (S.K.); heydari.yasaman712@gmail.com (Y.H.); 3Department of Radiology Medical Imaging Center, Imam Khomeini Hospital, Tehran 141-9733-141, Iran; Saeidmardpour@gmail.com; 4Department of Neurosurgery, Department of Applied Cell Sciences and Trauma Research Center, Kashan University of Medical Sciences, Kashan 87159-88141, Iran; 5Department of Immunology, Faculty of Medical Sciences, Tarbiat Modares University, Tehran 14115111, Iran; 6Faculty of Science, University of Toronto Scarborough, Toronto, ON M1C 1A4, Canada; amir.sharifzad@yahoo.com; 7Biotechnology Center, Silesian University of Technology, 44-100 Gliwice, Poland; mjelos@gmail.com; 8LinkoCare Life Sciences AB, 583 30 Linköping, Sweden; 9Faculty of Health, York University, Toronto, ON M3J 1P3, Canada; zahra.azizimd@gmail.com; 10Department of Human Anatomy and Cell Science, Max Rady College of Medicine, Rady Faculty of Health Sciences, University of Manitoba, Winnipeg, MB R3E 3P4, Canada; 11Faculty of Medicine, Katowice School of Technology, 40-555 Katowice, Poland; 12Pediatric Cell Therapy Research Center, Tehran University of Medical Sciences, Tehran 1419733151, Iran

**Keywords:** glioblastoma multiforme, rat model, NK cell therapy, MRI cell tracking, fluorescent cell tracking, blood brain barrier

## Abstract

Natural killer (NK) cell therapy is one of the most promising treatments for Glioblastoma Multiforme (GBM). However, this emerging technology is limited by the availability of sufficient numbers of fully functional cells. Here, we investigated the efficacy of NK cells that were expanded and treated by interleukin-2 (IL-2) and heat shock protein 70 (HSP70), both in vitro and in vivo. Proliferation and cytotoxicity assays were used to assess the functionality of NK cells in vitro, after which treated and naïve NK cells were administrated intracranially and systemically to compare the potential antitumor activities in our in vivo rat GBM models. In vitro assays provided strong evidence of NK cell efficacy against C6 tumor cells. In vivo tracking of NK cells showed efficient homing around and within the tumor site. Furthermore, significant amelioration of the tumor in rats treated with HSP70/Il-2-treated NK cells as compared to those subjected to nontreated NK cells, as confirmed by MRI, proved the efficacy of adoptive NK cell therapy. Moreover, results obtained with systemic injection confirmed migration of activated NK cells over the blood brain barrier and subsequent targeting of GBM tumor cells. Our data suggest that administration of HSP70/Il-2-treated NK cells may be a promising therapeutic approach to be considered in the treatment of GBM.

## 1. Introduction

Glioblastoma Multiforme (GBM) is among the most aggressive and lethal malignancies in humans [1]; it is a devastating brain disease, which typically results in mortality within 15 months postdiagnosis due to elevated intracranial pressure and interruption of normal functioning of brain tissues [2,3,4]. In addition, GBM grows fast and can spread quickly [5,6,7]. Localization of the tumor in the brain tends to aggravate its very aggressive nature, as it is out of the reach of the immune system; at the same time, brain cells have limited capacity to repair themselves [1,8]. Furthermore, leakage of the tumor capillaries causes fluid accumulation around the tumor and contributes to the aforementioned increase in intracranial pressure [2,8,9]. Current mainstay therapy for newly diagnosed GBM patients consists of a maximal resection within the safety limits; however, the tumor is usually infiltrative and complete resection is typically impossible, and resection must therefore be followed by radio- and chemotherapy [3,10]. Although tumors may respond to this approach, the likelihood of recurrence is extremely high due to the stemness property of this malignancy. Glioblastoma stem cells (GSCs), which constitute a small subpopulation of GBM cells with self-renewing and tumorigenic properties, contribute to disease initiation, its subsequent therapeutic resistance, and also have the potential ability to reconstitute the tumor after treatment [11,12]. Moreover, tumor heterogeneity, tumor location in a region where it is out of range of immune system control, and rapid tumor relapse represent additional main challenges complicating the treatment of GBM [10,13].

NK cells appear to be the most promising innate immune components when considering immune cell-based therapy treatment of GBM and other cancers, as they can recognize and kill tumor cells without any prior priming with tumor-specific antigens [14]. Aside from cytolysis, NK cells produce an array of cytokines, such as tumour necrosis factor (TNF) and interferons, which orchestrate immune responses [15]. Importantly, it has been postulated that NK cells target cancer stem cells [16]; however, the inadequate number and inactivity of NK cells at the tumor site are major concerns. To address these issues, a number of attempts have been made, most of which by manipulating the receptors on NK cells, via blocking inhibitory or activating stimulatory receptors (which both result in a net increase in the release of specific cytokines that suppress NK cell tolerance beyond expansion), or by engineering chimeric antigen receptors (CARs) to target tumor proteins [17,18].

Heat shock protein 70 (HSP70) is a 70 kilo Dalton (KD) chaperone protein, which is overexpressed in a number of tumor cells in various cancers, including GBM [19,20,21]. It provides a cytoprotective role and leads to inhibition of apoptosis in stressed cells. Elevation of HSP70 in cancer cells may be responsible for tumor progression by providing resistance to radio- and chemotherapy [22,23], and subsequent post-therapy survival [20,24,25]. Interestingly, it also serves as a target protein for activated NK cells [21,22,26,27], which recognize HSP70 as a tumor marker [26,28]. NK cells bind to HSP70 through the killer cell lectin-like receptor D1 (KLRD1), resulting in granzyme B release and tumor cytolysis [22,29].

In the present study, we have investigated the immunophenotype and proliferative capacity of IL-2/IL-15 cytokine-treated NK cells and their reactivity against green fluorescent protein (GFP)-labeled C6 cells in vitro. Subsequently, the potential cytolytic antitumor effects of IL-2/HSP70-treated NK cells were assessed in vivo in our rat models of GBM.

## 2. Results

### 2.1. Ex Vivo Expansion and Immunophenotyping of NK Cells

In order to expand NK cells, freshly isolated CD3^−^CD161^+^ cells were cultured in the presence of cytokines (IL-2+IL-15) and K562 feeder cells or IL-2+IL-15 alone. The resulting cells were expanded for over 16 days. Expanded NK cells exhibited round shape morphology with clonal growth (red arrow) under both culture conditions (Appendix A). In the presence of cytokines or cytokines/feeder cells, cells expanded 30- to 37.6-fold, respectively. The growth rate and yield of the cells were similar without noticeable differences between cytokine- and cytokine/feeder-treated cells for up to 16 days (Appendix A). Expanded cells were evaluated for the expression of NK cell surface markers; more than 90% of the population was CD3 negative and CD161 positive (Appendix A). Proliferation of cytokine-treated NK cells was also examined in co-culture with GBM tumor cells at different ratios. We found that carboxy fluorescein succinimidyl ester (CFSE)-labeled NK cells demonstrated a significantly higher proliferation rate at a 1:3 Target (T):Effector (E) cell ratio as compared to other ratios studied (Appendix A). It seemed that NK cells had high demand for nutrients to generate ATP, and biosynthesis of cellular macromolecules including lipids, proteins and nucleic acids. Since the level of essential nutrients such as glucose, fatty acid and glutamine were limited in culture media, this might cause a competition between effector cells. It could be an explanation for lower cytotoxicity and proliferation of NK cells [30].

### 2.2. In Vitro Cytotoxic Activity of Cytokine-Treated NK Cells

To assess cytotoxic activity of cytokine-treated NK cells, we co-cultured IL-2/IL-15-treated NK cells (E) and green fluorescent protein (GFP)-labeled GBM tumor cells (T) at different ratios. Flow cytometry analysis was performed through gating the GFP-positive/propidium iodide (PI)-positive cells as necrotic glioblastoma cells (Appendix A). Our results indicate that NK cells were cytotoxic towards GBM tumor cells. We also assessed the co-culture behavior of different ratios of T:E cells (1:1, 1:3, 1:5) (Appendix A). We found that there were no significant differences between different ratios. Interestingly, lactate dehydrogenase (LDH) activity in the supernatant, a cell lysis and necrosis biomarker, increased significantly as a consequence of co-culture with NK cells in comparison with the negative control (C6). However, no significant differences in the level of cytotoxicity were observed between the various concentrations T:E-ratios of NK cells studied. Overall, these results demonstrate the cytotoxicity of expanded NK cells (Appendix A).

### 2.3. Bio-Distribution of Intracranially and Systemically Injected NK Cells in a Rat GBM Model

Fluorescence in vivo imaging was used to evaluate the bio-distribution of cytokine-treated NK cells, 2 days after systemic injection in our induced GBM rat model. Paul Karl Horan (PKH-26)-labeled and IL-2/HSP70-treated NK cells demonstrated a potent ability to target GFP-expressing GBM cells within the tumor. Two days after injection, we observed the infiltration of IL-2/HSP70-treated NK cells around the GBM cells. Acquired images revealed a reduction in GBM tumor burden on day 8 (Figure 1). Figure 2A depicts the whole image of thick slices of brain tissue (350 µm) with tumor and NK cell injection sites in 4′,6-diamidino-2-phenylindole (DAPI)-stained sections as captured by fluorescent microscopy. For further confirmation, the animals were sacrificed, and brain tissues were delipidated to be optically transparent. 3D confocal imaging of clarified brain slices revealed precise targeting of GFP-expressing GBM tumor cells by PKH26-labeled NK cells (Figure 2B). Reconstructed images using Imaris software also confirmed the presence of NK cells in the proximity of GBM tumor cells (Figure 2C). These results demonstrate that NK cells can deeply penetrate into the GBM tumor and launch a pitched battle.

Systemic injection of NK cells was an approach used to assess whether treated NK cells were able to target tumor cells, even after traveling the longest distance from tail to head. Our data showed that the infiltration of NK cells inside the tumor tissue may correspond to their anti-tumor activity after systemic injection. Magnetic resonance imaging (MRI) molecular tracing was applied to determine homing of NK cells and their interaction with tumor cells. Superparamagnetic iron oxide (SPIO)-labeled NK cells were intracranially injected and MRI was performed to investigate NK cell localization; the details of this imaging protocol are described in our previous work [5]. MRI results demonstrated the infiltration of SPIO-labeled NK cells both around and inside the tumor area 2 days after intracranial injection. Furthermore, SPIO-labeled NK cells exhibited anti-GBM activity 8 days post intracranial injection (Figure 3).

### 2.4. In Vivo Functional Anti-GBM Activity of Rat NK Cells

To evaluate tumor establishment after C6 injection, all groups were observed for any changes in behavior, appetite, and body weight; MRI was performed to determine the tumor area 8 days after GBM tumor cell injection. Tumor mass, with necrotic area and edema as hallmark features of GBM, was observed in the MRI images.

In order to investigate the anti-tumor effect of NK cells, GBM-induced animals were divided into two trial groups: intracranial and systemic cell therapy models, which received IL-2/HSP70-treated NK cells or nontreated NK cells. MRI results revealed that as compared to the initial tumor size before treatment, tumor size was significantly reduced in the group intracranially injected with IL-2/HSP70-treated NK cells. However, no significant changes in tumor volume were observed after injection of nontreated NK cells (Figure 4A). The data showed a significant reduction in tumor size in animals exposed to treated NK cells versus those injected with nontreated NK cells. Furthermore, the survival rate was markedly improved in animals that had received treated NK cells compared to those subjected to nontreated NK cells. Animals treated with treated NK cells were alive for at least 90 days. Details of the in vivo anti-GBM effects of NK cells in the intracranial group are provided in Figure 4B.

MRI results showed similar antitumor activity, with markedly high tumor shrinkage and survival rate, of systemically administered treated NK cells (Figure 5A,B). Systemic injection of nontreated NK cells led to early death of the experimental animals.

Histopathological analysis of H&E-stained sections showed that all animals in the sham group developed a GBM tumor, characterized by pseudopalisading necrosis, microvascular hyperplasia, hypercellularity and pyknotic cells (Figure 6A). Importantly, animals in the systemic and intracranial groups that had received treated NK cells showed marked reduction of necrosis and microvascular hyperplasia 30 days after treatment as compared to animals in the sham group (Figure 6B,C). As presented, we also observed infiltration of lymphocytes in both groups injected with treated NK cells [31,32].

## 3. Discussion

Extensive genetic and cellular diversities render GBM difficult to treat and eradicate. In spite of various modalities of treatments, overall survival has only modestly increased over the last 30 years [3,8,33,34]. Immune cell therapy has emerged as a promising tool to tackle GBM [35,36,37,38]. Several studies demonstrated immune cell infiltration in the brain of patients with malignant glioma, although it was previously considered as an immune-privileged organ [39,40]. However, tumor-derived factors suppressed immune responses of these cells [41]. Previous studies showed that tumor-residing glioma cells regulate the immune response through recruitment of immune-suppressive cells, which represents a challenge in the development of cell immunotherapies [42].

NK cells have been considered as an effective treatment modality for cancer treatment for over 50 years; however, efficacy has been limited due to the availability of relatively small numbers and lack of sufficient activation in the vicinity of the tumor [43]. Such concerns are even more important in GBM due to the unique localization of the tumor, which limits trafficking of NK cells over the blood-brain barrier (BBB) from the blood into the brain [44]. In the first step of our study, evaluation of NK cell cytotoxicity and proliferation demonstrates the NK cell effectiveness against GBM cells (C6) *in vitro*. Although the NK cell/C6 cell co-culturing conditions are not fully identical to the environment in vivo, this in vitro setup serves as an optimally controllable system to closely predict the effects of NK cells on tumor cells *in vivo*.

Results of the CFSE proliferation assay proved rapid expansion of NK cells in co-culture with the GBM cell line. Moreover, in vitro cytotoxicity assessment indicated the lysis potential on C6 tumor cells. The PI incorporation test and LDH release assay were used to determine the cytotoxic effects [40]; these methods were chosen over assessing chromium-51 release out of ecologic considerations. For the PI incorporation assay, we co-cultured NK cells with GFP-labeled C6 cells and assessed the rate of C6 cell death by selecting GFP/PI positive cells among total GFP-expressing cells, using flow cytometry. This protocol is cost-effective and, unlike the chromium-51 release assay, does not produce hazardous waste. Flow cytometry data indicated a cytotoxicity of stimulated NK cells on C6 cells, which was more pronounced with a higher number of NK cells present (dose-dependent results). The LDH assay further confirmed the cytotoxic capacity of activated NK cells, albeit with an apparent lower sensitivity, as significant differences between various doses of NK cells could not be identified, suggesting dose-dependency is subtle and might only be detected with more sensitive approaches.

Based on the results obtained from our in vitro experiments, we evaluated the antitumor effects of *ex vivo*-expanded and HSP70/IL-2-treated NK cells in our GBM rat models and compared two different routes of administration of NK cells (systemic versus intracranial).

In the in vivo part of our studies, we aimed to address key questions such as (i) is HSP70\IL-2 treatment sufficient to facilitate recruitment of NK cells into the tumor site? and (ii) is this stimulation robust enough to enable sufficient numbers of activated NK cells cross the blood-brain barrier and trigger the antitumor activity? In our previous study, we investigated the similarities between human GBM and the C6-induced GBM rat model [5]. By applying bioinformatic tools, MRI and histopathology assessments, we established that this model closely resembles the human condition, to an extent that it could be exploited in preclinical immune cell therapy assessment. Our current data demonstrate that IL-2/HSP70 can effectively stimulate NK cells, enhancing their targeting, homing and cytolytic properties. We implemented several different protocols: (a) live in vivo imaging, which showed the accumulation of NK cells around the tumor microenvironment, and tumor shrinkage 8 days after NK cell injection; (b) 3D imaging by confocal microscopy of thick brain slices, which confirmed tumor targeting by PKH-26-labeled NK cells. In addition, deep penetration of PKH-26-labeled NK cells into the GBM tumor was evident, supporting the ‘death kiss’ hypothesis, which is a symbolic expression for NK cell-induced cytolysis, as previously described [45,46]; and (c) MRI tracing of SPIO-labeled NK cells, which demonstrated tumor targeting after 48 h and tumor shrinkage 8 days after SPIO-labeled NK cell therapy. To the best of our knowledge, this is the first preclinical study showing the tug of war between NK and tumor cells on a cellular and molecular level.

Finally, the antitumor activities of naïve nontreated and IL-2/HSP70-treated NK cells were evaluated and compared by MRI, showing effectiveness of treated NK cells manifested by tumor shrinkage, both after intracranial and systemic injection. Of note, the tumor size in rats treated with systemically injected, nontreated NK cells could not be properly evaluated due to the early, tumor-related death of those animals. Intravenous injection of IL-2/HSP70-treated NK cells, which targeted GBM cells in the brain, did further confirm the effectiveness of activated NK cells, which exhibited specific homing properties, accumulating in the brain, and reaching the tumor cells. It is noteworthy to emphasize that, compared to naive nontreated NK cells, a short-term treatment with HSP70/IL-2-activated NK-cells results in significant tumor shrinkage and survival.

A small number of studies exist in which attempts were made to enhance the cytotoxic effect of NK cells towards tumor cells. For instance, the utilization of a programmed cell death protein (PD-1) blocker in a multiple myeloma model, resulted in increased NK cytotoxicity markers and therapeutic efficacy toward this malignancy [47]. A follow-up study demonstrated the cytotoxicity enhancement of TKD (antigenic component of HP70)/IL-2-activated NK cells by PD-1 blocker stimulation, which resulted in local tumor control in immunodeficient and/or immunocompetent mice, in GBM and lung cancer models [48]. However, the expression of PD-1 on GBM cells is highly questionable [49]. On the other hand, GBM represents an immunosuppressive microenvironment characterized by low immunogenic responses. Therefore, whether these results are related to PD-1, TKD/IL-2 or a synergistic or additive effect of both stimulation methods remains to be elucidated.

The results of our in vivo studies provide, at least to some extent, answers to the aforementioned questions, since the anatomical size of the rat is large enough to evaluate the tumor after immune cell therapy. Furthermore, many genetic and signaling pathway perturbations, and key pathological features that can be evaluated by MRI, are common between autochthonous human GBM and induced rat GBM according to our previous studies; therefore, we are confident that data generated by our experimental in vitro and in vivo systems are of clinical importance and open the door for further translational research.

## 4. Material and Methods

### 4.1. Cell Line

The rat glioblastoma cell line (C6) was purchased from the National Cell Bank of Iran (NCBI), Pasteur institute, C575. Cells were cultured in DMEM F-12, supplemented with 2.5% fetal bovine serum and 12.5% horse serum.

Cells were observed daily and, after formation of a confluent monolayer, passaged serially. Expanded in such way, cells were divided into two groups: (1) cells that were frozen for future use, and (2) cells that were labelled with GFP for tracing experiments in the (GBM) rat brain, as previously described [50].

### 4.2. Natural Killer Cell Isolation, Expansion, and Stimulation

All animal protocols have been approved by the Review Board and Institutional Ethics Committee of the Royan Institute (IR.ACERCR.Royan.REC.1395.59 21/06/2016). Mononuclear cells (MNCs) were obtained from rat spleens, followed by density separation using Lympholyte^®^ Cell Separation density gradient centrifugation media (Cedarlane, Canada). These MNCs were subjected to FACS positive selection of NK cells. Next, NK cells were isolated by positive selection of CD161+CD3- cells using FACS. Briefly, MNCs were stained using phycoerythrin (PE)-conjugated anti-CD161, and fluorescein isothiocyanate (FITC)-conjugated anti-CD3, according to the manufacturer’s protocol.

To obtain optimal NK cell expansion, we implemented two different protocols and compared their results. Isolated NK cells were cultivated in complete RPMI 1640 medium, with different enrichment approaches: the first group was supplemented with K562 as feeder cells and IL-2 (1000 IU/mL) and IL-15 (100 IU/mL), the second group was supplemented with cytokines alone. Both groups were incubated in a humidified incubator (5% CO2 at 37 °C).

In accordance with the previously published procedure, NK cells were activated by adding HSP70 (2 µg/mL) and IL-2 (100 IU/mL) to the culture medium, and further incubated under standard cell culture conditions for four days [29].

### 4.3. In Vitro Proliferation Assay, NK Cells In Co-Culture with GBM Cells

To probe NK cell proliferation in the vicinity of the C6 cell line, isolated NK cells were labeled with CFSE through a previously established method with a few modifications [51,52]. Briefly, NK cells were dyed with 1 μL CFSE and cultured in complete RPMI 1640 medium, in a humidified incubator (5% CO2, at 37 °C) for 72 h.

### 4.4. NK Cells in Vitro Cytotoxicity Assays

The cytotoxic effects of NK cells on C6 tumor cells were determined in a co-culture of GFP-labeled C6 (target, T) cells and NK (effector, E) cells; T:E: ratios of 1:1, 1:3, and 1:5 were studied. After 24 h of incubation, the supernatant was collected for an LDH assay, as a necrosis marker in cell culture medium [53], which was performed using a kinetic method, and cells were analyzed using a FACS-CANTO flow cytometer. To determine the relative extent of apoptotic and necrotic cell death, propidium iodide (PI; 1 µg/mL) was added.

### 4.5. Animal Studies

#### Glioblastoma Rat Modeling

Animal studies were performed using a total of 14 male rats with body weights of 230–250 g. All animal experiments were approved by the review board and institutional ethics committee of the Royan Institute (IR.ACERCR.Royan.REC.1395.59 21/06/2016). Selected animals were anesthetized with ketamine/xylazine (55/11) mg/kg by peritoneal injection, and subjected to stereotaxic surgery, according to our previously published protocol [5]. Briefly, animals were divided into two groups: the experimental group receiving C6 glioblastoma cells, and a sham-exposed group receiving culture media. Any changes in movement ability and bloody discharge of the eyes were noted as tumor implantation signs. Tumor establishment and treatment follow-up was assessed by MRI [5].

### 4.6. NK Cells Bio-Distribution Assays

We implemented three different procedures for in vivo tracking of NK cells:

(a) MRI molecular tracing: NK cells were labeled with commercially available (SPIO) nanoparticles at a concentration of 100 μg/mL with protamine sulfate as a transfecting agent at a final concentration of 45 μg/mL. The mixture was added to FBS/l-glutamine free culture media containing NK cells for 2 h. Subsequently, we added 1% l-glutamine and 10% FBS and incubated for 48 h. After this period, these labeled cells were washed twice with sterile PBS and introduced to GBM-induced rats by intracranial injection, followed by tracking by MRI [54,55]. MRI was performed 48 h post injection with SPIO-labeled NK cells using a 3.0 T Prisma, Siemens apparatus equipped with a 75-rat coil. T2 mapping with the acquisition of T1- and T2-weighted anatomical images was performed.

(b) Live in vivo fluorescent imaging: (PKH)-26 staining (emitted at 567 nm) was utilized in an additional attempt to trace NK cells and identify their potential interactions with tumor cells. For this purpose, NK cells were centrifuged for 5 min at 400× *g*, followed by suspension of the pellet in 1 mL diluent C to prevent direct contact of cells with dye. In a separate vial, a concentration of 4×10^−6^ mol/L was prepared by diluting 4 μL PKH-26 red fluorescent dye in 1 mL diluent C, prior to cell labeling. Suspended NK cells were mixed with diluted PKH-26 dye and incubated for 5 min at room temperature, gently inverted to disperse, and finally washed with 1% BSA PBS to stop the staining process. These cells were used for systemic injection. An additional batch of tumor cells, planned for implantation, were transfected with GFP (emitted at 509 nm), which allowed us to monitor them by detecting two distinct wavelengths using a Kodak in vivo F series imaging system (New Haven, CT, USA).

(c) 3-Dimensional brain imaging using fast Free-of-Acrylamide Clearing Tissue (FACT): We utilized fast Free-of-Acrylamide Clearing Tissue (FACT), which has been recently established [56], as a third protocol to study NK cells *in vivo*. Herein, we employed this tissue clearing method to achieve fluorescent three-dimensional imaging of rat brain tissue. To improve the efficacy, we perfused animals with normal saline followed by 4% paraformaldehyde prior to decapitation. One mm thick slices of brain tissue were subjected to clearing by 8% SDS (pH: 7.5) at 37 °C for 42 h, to completely delipidate the brain section. Slices were subsequently washed with PBS for 12 h to remove the excess SDS. To improve clarity and transparency, brain slices were incubated with sorbitol for 4 h. Next, slices were subjected to confocal microscopy (Concord, ON, Canada) (Leica; Z size: 330 µm). The Imaris surface algorithm was applied to create a 3-dimensional reconstruction of GFP-expressing GBM cells and PKH-labeled NK cells.

### 4.7. Histopathology Assessments

Animals were sacrificed through perfusion (to achieve proper fixation and preserve brain tissue from any damage until time of experimentation [57]). Tissues were subjected to paraffin embedding; subsequently, blocks were sectioned at 5 µm thickness by a microtome. Hematoxylin & Eosin (H&E) was applied for general histopathological studies.

### 4.8. Statistics

Statistical analysis was performed using GraphPad Prism software (version 6) and data are presented as mean ± SD from at least three independent experiments. We utilized one way analysis of variance (ANOVA) followed by a Tukey (or Holm-Sidak for MRI results) post hoc test to evaluate the statistical significance of differences between groups. Differences were considered to be statistically significant when *p* < 0.05.

## 5. Exciting Prospects and Existing Questions

Considering our findings on the antitumor effects of HSP70/IL-2-treated NK cells in our in vivo rat model of induced GBM, we envision translation of this novel treatment approach to tackle GBM in humans, although there are some ambiguities which remain. Our studies did not directly shed light on mechanistic processes underpinning NK cell activities. Further, in depth studies are warranted to identify the exact mechanism that drives NK cell recruitment through the BBB towards the tumor site. In this respect, two distinct agents appear to be involved: HSP70 and neurotactin (CX3CL1 or fractalkine) [58,59,60]. The interactions between neurotactin/fractalkine and HSP70 likely play a key role in the chemoattraction towards and crossing over the BBB. Regarding our studies, one of the most important questions remaining is if the activation of HSP70 with IL-2 is of a synergistic or a central nature. The microenvironment components and their effects on immune cells as well as their mechanisms of action are other important issues to be addressed. Moreover, it would be valuable to determine if stimulation of NK cells with HSP70/IL-2 could induce a memory of NK cells to prevent future tumor recurrence. Taken together, our findings have established that ex vivo HSP70/IL-2 stimulation potently activates NK cells, which enables them to effectively cross the BBB and target tumor cells in our in vivo rat model of induced GBM.

## Figures and Tables

**Figure 1 ijms-21-02263-f001:**
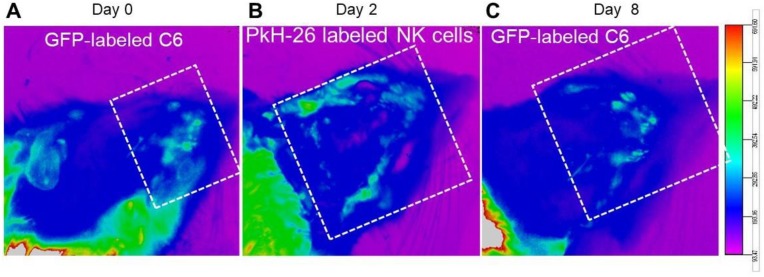
Bio-distribution of NK cells in the rat model of induced GBM; live imaging. Bio-distribution of PKH-26-labeled NK cells 2 days after intravenous injection in our GFP-labeled GBM tumor induced model. (**A**) GFP-labeled GBM cells were found at day 0 (tumor establishment, white dotted frame). (**B**) Fluorescence imaging on day 2 shows the localization of PKH-labeled NK cells around GFP-labeled tumor cells (white dotted frame). (**C**) A reduction of fluorescence intensity of GFP-labeled GBM cells was evident 8 days post injection, indicating the effectiveness of the introduced NK cells (white dotted frame).

**Figure 2 ijms-21-02263-f002:**
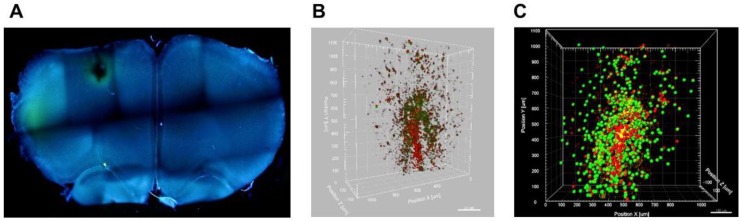
NK cell homing and interaction with tumor cells. (**A**) Representative image of a thick brain section stained with DAPI to identify the tumor and NK cell interaction zone. (**B**) 3-Dimentional confocal imaging showing deep penetration of PKH-labeled NK cells into the tumor (GFP-labeled tumor cells). (**C**) An overview of NK cells (red spheres) and tumor cells (green spheres) as reconstructed by Imaris software.

**Figure 3 ijms-21-02263-f003:**
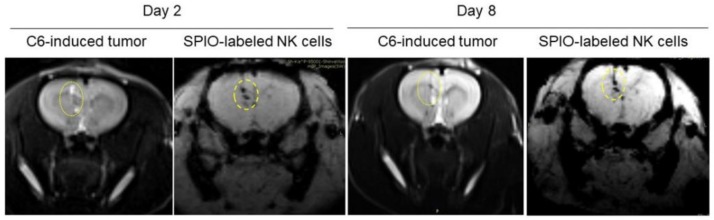
NK cell tracing by MRI. MR imaging of SPIO-labeled NK cells 2 and 8 days after intracranial injection. Yellow ellipses point at approximate location of the injected GBM cells.

**Figure 4 ijms-21-02263-f004:**
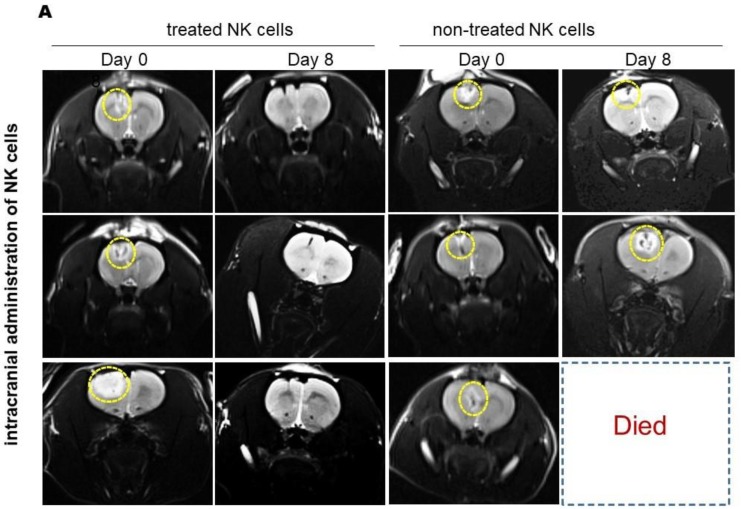
MRI results of intracranial injection of NK cells in the rat model of induced GBM. (**A**) T2-weighted MR Images of GBM tumor-bearing rats show tumor shrinkage over time to nearly undetectable size at day 8, after injection with HSP70/IL-2-treated NK cells (panels on the left). No robust effects of injection with nontreated NK cells were observed (panels on the right) yellow dotted circles show tumor zones. (**B**) Graph depicting tumor volume shrinkage at day 8 in rats intracranially injected with treated NK cells versus nontreated NK cells (** *p*-value: 0.0319). Tumor size in missed (dead) cases was considered as unchanged.

**Figure 5 ijms-21-02263-f005:**
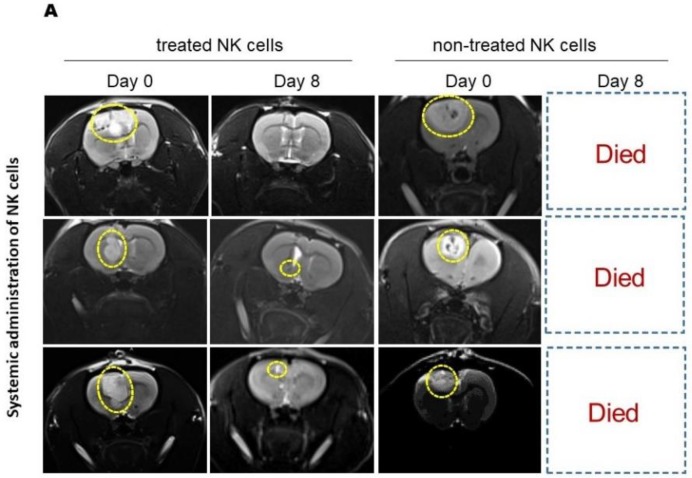
MRI analysis of the effects of systemically administrated NK cells on induced GBM in rats. (**A**) T2-weighted images (acquired by 3T MRI) revealed reduction of GBM tumor volume 8 days after systemic injection of IL-2/HSP70-treated NK cells (panels on the left). Animals that received nontreated NK cells did not survive until day 8 (panels on the right) (yellow dotted circles show tumor zones). (**B**) Graph depicting tumor volume shrinkage at day 8 in rats systemically injected with treated NK cells versus nontreated NK cells (Holm–Sidak test, adjusted * *p*-value: 0.0029). Tumor size in missed (dead) cases was considered as unchanged.

**Figure 6 ijms-21-02263-f006:**
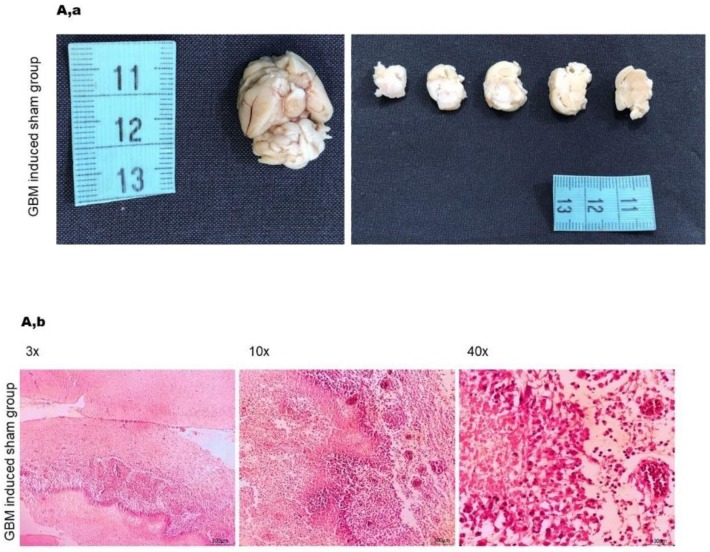
Histopathological analysis of brain sections following injection of HSP70/IL-2-treated or nontreated NK cells. (**A,a**) Gross image morphology of sham group, cerebral lesion confirmed microscopically to be GBM. (**A,b**) Histopathological images acquired after H&E staining illustrate palisading necrosis, nuclear pleomorphism with atypia and cell debris, which confirmed GBM; the normal brain tissue at the top of the picture is compared to the GBM tissue (left picture). Palisading necrosis with vascular proliferation, marked nuclear pleomorphism, atypia, brisk (TIL infiltrations around the tumor: rimming) mitosis and karyorrhexis: (middle picture, low magnification). High power field images show palisading necrosis with microvascular proliferation, marked nuclear pleomorphism with atypia, brisk mitosis and karyorrhexis: GBM in the sham group (right picture). (**B,a**) Gross morphology of the rat brain and cerebellum of the group that received intracranial injection of IL-2/HSP70-treated NK cells represent smooth brain surface, no lesions are identified. (**B,b**) Images of the experimental group that received intracranial injection of IL-2/HSP70-treated NK cells indicate normal cellularity, normal glial cells, neurons and astrocytes in the neuropil background as compared to GBM control, Figure 6A,b. No signs of necrosis, inflammation or desmoplasia were observed (30 days post injection). (**C,a**) Gross morphology of the group that received systemic injection of IL-2/HSP70-treated NK cells, the rat brain and cerebellum represent smooth brain surface, no lesions are identified. (**C,b**) Images of the experimental group that received systemic injection of IL-2/HSP70-treated NK cells indicate normal glial cells, neurons and astrocytes in the neuropil background with mild lymphocytic infiltration, and few visible capillaries as compared to GBM control, Figure 6A,b (30 days post injection). (**D,a**) Gross morphology of the rat brain and cerebellum of the group that received intracranial injection of nontreated NK cells. Smooth brain surface. No lesions are identified. (**D,b**) Images of the experimental group that received intracranial injection of nontreated NK cells showed normal glial cells, neurons and few atypical astrocytes with focal lymphocytic infiltration in a neuropil background compared to GBM positive control, Figure 6A,b (30 days post injection).

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
