# Peer review of "HSP70/IL-2 Treated NK Cells Effectively Cross the Blood Brain Barrier and Target Tumor Cells in a Rat Model of Induced Glioblastoma Multiforme (GBM)"

_ijms, 2020, doi:10.3390/ijms21072263_

Round 1
Reviewer 1 Report
This is a very wll written and comprehensive manuscript. Authors investigated the cytotoxic potential of IL-2/IL-15 cytokine treated NK cells against GFP-C6 79 cells in vitro. Moreover, cytotoxicity of NK-treated cells was also tested in vivo in rat models of GBM. Authors showed that NK-treated cells when administered either systemically or intracranially exert a significant cytolytic activity against GBM, resulting in a significant prolongation in survival. On the contrary no effect was observed after the administration on non-treated NK cells. Results are very interesting
Below there is a list of suggestions for further improving the quality of the manuscript
1) In section 2.1: When NK cells were co-cultured in the presence of C6 cells then the optimal NK expansion occurred at a target:effector ratio of 1:3. Authors should comment why NK expansion was reduced at increasing target cell ratios
2) In section 2.2: Authors claimed that they found that a 1:3 ratio exhibited a trend towards strong lytic activity on GBM cells compared to other cell ratios, although there were no significant differences between different ratios. This sentence should be deleted because its misleading. As is shown in supplementary Figure 4 there was no difference in cytotoxic activity at various T:E ratios. Moreover there is no p value written and the phrase ''there is a trend'' is not correct
3) In supplementary Figure 4: Figure 4A is missing
Author Response
Comment 1: When NK cells were co-cultured in the presence of C6 cells then the optimal NK expansion occurred at a target:effector ratio of 1:3. Authors should comment why NK expansion was reduced at increasing target cell ratios
Answer: We appreciate the respected reviewer attention and comment. Actually, we claimed that, NK cells expansion reduced. We suggested that the amount of nutrient resources in culture media was probably the main reason for reduction of NK cells numbers. It seems that all lymphocyte subsets including NK cells has different demands for nutrients to generate ATP, and to biosynthesis of cellular Macromolecules including lipids, proteins and nucleic acids. Since the level of essential nutrients such as glucose, fatty acid and glutamine are limited in culture media; this could be an explanation for lower cytotoxicity and proliferation of NK cells in greater amounts (Line 96-100).
Comment 2: Authors claimed that they found that a 1:3 ratio exhibited a trend towards strong lytic activity on GBM cells compared to other cell ratios, although there were no significant differences between different ratios. This sentence should be deleted because its misleading. As is shown in supplementary Figure 4 there was no difference in cytotoxic activity at various T:E ratios. Moreover there is no p value written and the phrase ''there is a trend'' is not correct
Answer: We appreciate the respected reviewer comments. We have applied the comment and deleted aforementioned sentence (Line 107-108).
Comment 3: In supplementary Figure 4: Figure 4A is missing
Answer: We have added the missing panel in the revised version of the supplementary figures
Reviewer 2 Report
In my opinion the manuscript entitled “ HSP70/IL-2 Treated NK Cells Effectively Cross the 2 Blood Brain Barrier and Target Tumor Cells in a Rat 3 Model of Induced Glioblastoma Multiforme (GBM)” is very interesting and could have an important applicative potential for the cancer care.
The manuscript is well written and organized. Leading edge techniques and tools have been used for the experiments.
I suggest only minor revision to the authors.
The authors should add photo of excised tumors to demonstrate effectively the size of tumors in the different group of rat.
Some more recent citations should be added. Moreover, the references should be updated following the journal’ style.
Author Response
Comment 1: The authors should add photo of excised tumors to demonstrate effectively the size of tumors in the different group of rat.
Answer: The authors appreciate the respected reviewer for the important comment. We have added excised tumors images (lines: 200,204,206,208) and added the corresponding legends to the Figure legends (lines 211-212, 219-220, 223-225, 229-230, and 233) in the revised manuscript,
Line 202: Figure 6A,a Cerebral lesion confirmed microscopically to be GBM.
Comment 2: Some more recent citations should be added. Moreover, the references should be updated following the journal’ style.
Answer: We have revised the old referecnes and have updated them. They are in lines: 62,100,240, and updated the following the journal style.
This manuscript is a resubmission of an earlier submission. The following is a list of the peer review reports and author responses from that submission.